# Influences of Animal-Assisted Therapy on Episodic Memory in Patients with Acquired Brain Injuries

**DOI:** 10.3390/ijerph17228466

**Published:** 2020-11-16

**Authors:** Felicitas Theis, Frank Luck, Margret Hund-Georgiadis, Karin Hediger

**Affiliations:** 1Clinical Psychology and Psychotherapy, Faculty of Psychology, University of Basel, Missionsstrasse 62a, 4055 Basel, Switzerland; karin.hediger@unibas.ch; 2Gender Studies, University of Basel, Rheinsprung 21, 4051 Basel, Switzerland; frank.luck@unibas.ch; 3REHAB Basel, Clinic for neurorehabilitation and paraplegiology, Im Burgfelderhof 49, 4055 Basel, Switzerland; m.hund@rehab.ch; 4Swiss Tropical- and Public Health Institute, Department of Epidemiology and Public Health, Socinstrasse 57, 4051 Basel, Switzerland

**Keywords:** animal-assisted therapy 1, acquired brain injury 2, neurorehabilitation 3, episodic memory 4

## Abstract

Animal-assisted therapy (AAT) is shown to be an effective method to foster neurorehabilitation. However, no studies investigate long-term effects of AAT in patients with acquired brain injuries. Therefore, the aim of this pilot study was to investigate if and how AAT affects long-term episodic memory using a mixed-method approach. Eight patients rated pictures of therapy sessions with and without animals that they attended two years ago. Wilcoxon tests calculated differences in patients’ memory and experienced emotions between therapy sessions with or without animals. We also analyzed interviews of six of these patients with qualitative content analysis according to Mayring. Patients remembered therapy sessions in the presence of an animal significantly better and rated them as more positive compared to standard therapy sessions without animals (Z = −3.21, *p* = 0.002, g = 0.70; Z = −2.75, *p* = 0.006, g = 0.96). Qualitative data analysis resulted in a total of 23 categories. The most frequently addressed categories were “Positive emotions regarding animals” and “Good memory of animals”. This pilot study provides first evidence that AAT might enhance episodic memory via positive emotions in patients with acquired brain injury.

## 1. Introduction

Learning is a central factor for neurorehabilitation of patients with acquired brain injury [1,2,3]. Considering the treatment costs for one single patient with acquired brain injury and the often-resulting long-term consequences, effective treatment approaches are highly important [4]. For a therapy to be effective in enabling learning, patients need to remember the content of a certain therapy session and need to be able to memorize content in the long-term. The neurobiological construct referring to this ability is the episodic memory. Episodic memory is especially crucial to attain sustainable effects of emotional based learning [5]. 

Research shows that learning and the memory ability are positively influenced by emotional cues [6]. Studies highlighted the association between emotions in a certain situation and an enhanced episodic memory [7,8], especially positive emotions in a situation seem to strengthen the episodic memory of this situation [9]. This effect is probably driven by a stronger encoding and is found for both positive and negative emotions [10]. It was shown that memory is especially enhanced by emotions driven by stimuli that have a high emotional relevance for us [11,12,13,14]. Humans develop close emotional bonds and relevant social relationships with animals [15,16]. Animals might be a stimulus that lead to stronger encoding, since they are emotionally meaningful for humans [17,18], and interacting with animals seems to promote positive emotions in different populations [19]. This mechanism might be an underlying process of the effects of animal-assisted therapy (AAT) found in neurorehabilitation. Research shows that AAT can lead to enhanced social behaviour, positive emotions and therapy motivation in patients with acquired brain injury [20]. In patients in a minimally conscious state, the presence of an animal can lead to more behavioural reactions and increased physiological arousal compared to control sessions [14]. However, there are no studies investigating long-term effects of AAT on memory in patients with acquired brain injuries.

The aim of this pilot study was to investigate long-term effects of AAT on episodic memory in patients with acquired brain injury who participated in a precursor study two years ago [14]. We hypothesized that patients remember therapy sessions in the presence of an animal better after two years than comparable standard therapy sessions and that this effect might be driven by increased positive emotions in the presence of an animal.

## 2. Materials and Methods 

### 2.1. Participants 

We included patients with acquired brain injury who participated in a study two years earlier [16]. Exclusion criteria were low cognitive or speech abilities as the patients needed to be able to complete the open interview or/and the written questionnaire. We tried to contact all 22 previously included patients by telephone or written communication. We were able to get in contact with 18 former patients of which 8 (6 males, 2 females) agreed to participate in this follow-up study. None of them met the exclusion criteria. Interestingly, more men than women participated in the study. This fact makes sense since more men than women participated in the previous study. The reason for this could be that most of the patients from the previous study were treated after a road accident. Epidemiological studies show that men are more likely to be victims of road accidents [21]. All participants were from regions of German-speaking Switzerland and were previously in inpatient neurorehabilitation due to an acquired brain injury. During data collection, all of the participants were no longer in an inpatient setting. The mean age of the sample was 55.8 years (sd = 6.7). The study protocol was approved by the Human Ethics Committee for Northwest and Central Switzerland, and all participants or their legal representatives provided written informed consent. 

### 2.2. Design and Procedure 

The study was designed as a follow-up pilot study with a mixed-methods approach. The quantitative part used a randomized, controlled within-subject design and the qualitative part consisted of semi-structured interviews. Each participant participated in a precursor study with a controlled within-subject design with repeated measurements 24 month prior to this study. In the precursor study, every participant received both AAT sessions and paralleled conventional therapy sessions over a six-week period [14]. In this way, each participant had previously received 24 therapy sessions, of which 12 sessions were performed in the presence and 12 sessions in absence of an animal. The sessions were matched within one patient for activities, goals and settings. Involved animals were guinea pigs, rabbits, chickens, miniature pigs, goats, sheep, horses, donkeys, cats and dogs. In AAT sessions, patients have been given tasks to care for or feed the animal whereas in the control sessions, similar tasks were carried out but without the presence of an animal [14]. The patients and the animals actively interacted during the AAT sessions but often for only part of the time of a whole session. The present study took place at REHAB Basel, the clinic where the patients attended neurorehabilitation before their release, either to their home, to a further rehabilitation center or to a nursing home. One appointment was performed at a participants’ home because the participant was not able to come to the clinic.

We chose to use a mixed-method approach to account for the possible reduced cognitive functions in the study sample. Acquired brain injuries can lead to memory impairments as well as impaired affective processes [21]. In contrast to quantitative methods, a qualitative approach allows for greater personalization to address such difficulties. The study procedure included an open interview including a maximum of 24 questions in a first step (approx. 30 min duration) and was followed by a questionnaire (approx. 15 min duration) in a second step.

### 2.3. Measures 

A quantitative questionnaire was designed to measure the degree to which patients were able to remember therapy sessions (strength of episodic memory) as well as arousal and valence of these remembered therapy sessions. The questionnaire had the same structure but with individualized items for every participant. Each questionnaire consisted of four individual pictures from past therapy sessions extracted from videotapes of the precursor study. Two pictures showed a therapy session without an animal, two pictures a paralleled therapy session with the same therapy conditions (showing the same therapist in a similar environment) but in the presence of an animal. Therefore, every participant evaluated two picture-pairs, resulting in a total of 16 rated picture-pairs over all 8 participants. Participants decided which therapy session of each picture pair was more memorable (“Which picture do you remember better?”) and rated each picture on an emotional scale (“How good or bad do you feel while looking at the picture?”, “How relaxed do you feel while looking at the picture?”). This design reflected the circumplex model of emotions consisting of a six-point Likert scale asking for arousal and valence (arousal: 1 = high positive emotions, 6 = high negative emotions; valence: 1 = low valence, 6 = high valence). Rating scales needed to be easily understood by patients. For this reason, common smileys from a messenger app were used [22], with smileys symbolizing positive/negative emotions and smileys symbolizing relaxed/high arousal state. To control for effects of different therapists and therapy contents, pictures of two different therapy forms (occupational therapy, speech therapy or physiotherapy) were used for each participant. The order of the different kinds of therapy sessions as well as the arrangement of the pictures with or without an animal in the questionnaire were randomized to control for priming and recency effects using an electronic random number generator [23]. The questionnaire was completed using paper and pencil. The dependent variable was the therapy condition: presence/absence of the animal. The independent variables were patients’ self-rated memory strength and patients’ self-rated emotional arousal/valence. The questionnaire can be found Appendix A.

The qualitative interview focused on the patients’ general memories of the rehabilitation stay, memories of AAT, the relationship between patients and the involved animals during therapy sessions as well as the patients’ opinion of whether AAT is effective. The interview was semi-structured, so that the question number and the order could be adapted individually to each participant. Due to the design, fixed variables could not be anticipated. Interview questions were developed by an expert group consisting of the investigator and the principal investigator of the study as well as an expert in qualitative data analysis. Only open questions were used to minimize bias through directed questions. We started with four general questions, not focusing on AAT, to assess if participants remembered AAT spontaneously. Patients were blind regarding the study aim and did not know that we wanted to investigate their memories about AAT. Afterwards, we asked the questions targeting the actual study topic. We finished the interview by asking for a personal summary of participant’s most important memories of the rehabilitation stay as well as the most important memories of AAT. The interview guideline is presented in Appendix B. We performed eight interviews from which we could include six for qualitative analysis. The reason for exclusion was the presence of family members during two interviews. 

### 2.4. Qualitative Content Analysis

The interview material was audio taped, transcribed and analyzed using the qualitative content analysis approach of Mayring [24]. We developed the category system for each interview inductively. Therefore, every text unit was paraphrased. Paraphrases were reduced and grouped into higher subcategories. In a next step, subcategories were generalized, reduced and integrated in one superior category system. Finally, interview material was reviewed on the basis of the developed category system. We directly counted how many times each category from the final category system was mentioned by the participants in each of the six interviews. The presented process was iteratively discussed by the expert group, as content analysis is not a one-sided procedure but rather repeats itself with the aim to reduce the interview material to the essentials [24]. 

### 2.5. Statistical Analysis

To investigate the effects on memory, we counted the frequency of AAT and control pictures that were rated by the participant as remembered better. For arousal and valence, we estimated the mean, median, standard deviation and 95-percent confidence interval for memory ability. After data preparation, we calculated Wilcoxon Tests to detect differences between therapy sessions in the presence or absence of an animal for the three quantitative outcomes (memory ability, arousal and valence). We performed quantitative data analysis for all eight participants. One participant refused to appraise one of the picture pairs leading to a final data set of 15 analyzed pairs of pictures representing 30 therapy sessions. We designated a *p*-level of 0.05 to be statistically significant. We calculated effect sizes as Hedges g [25], with the interpretation of 0.8 as large, 0.5 as moderate and 0.2 as small effect [26]. We performed the statistical data analysis in RStudio (3.5.0). We analyzed quantification of the mentioned categories during the interviews using descriptive statistics. 

## 3. Results

### 3.1. Quantitative Measures 

The participants remembered the therapy session in the presence of an animal better in 12 out of 15 picture pairs. This outcome was statistically significant with a moderate effect size (Z = −3.21, *p* = 0.002, g = 0.70). Moreover, patients rated the pictures from the AAT sessions with significant higher values for positive emotions (Z = −2.75, *p* = 0.006, g = 0.90) and lower values in arousal compared to pictures from standard therapy sessions (Z= −3.36, *p* = 0.03, g = 0.96), with high effect sizes for both parameters. Table 1 shows the related statistical parameters. 

### 3.2. Qualitative Interview 

Qualitative data analysis resulted in a total of 23 categories, of which 10 were related to our research question. Table 2 shows the full category system. The table shows results for the first 10 categories about how often each category was mentioned in every single interview and in total. The further 13 categories were not counted out because they were not AAT specific and/or not related to our research question. 

The most frequently addressed categories were category 1 “Positive emotions regarding animals” and category 2 “Good memory of animals”. An example of category 2 was a participant who reported remembering animals and events with animals was easier for him than remembering events with humans. Within category 1, participants characterized AAT sessions by a multitude of positive emotions. They viewed animals as a source of joy and pleasure. A few participants remembered moments as funny, when the animal acted in an unexpected way which made them laugh. Apart from this, animals were seen as a transmitter of unique calmness and emotional consistency. For example, one participant especially enjoyed the contact with animals because they were not moody. Frequently, these qualities were set in contrast to healthcare professionals, with whom participants had an unpleasant experience in their past.

Category 3, the statement “a relationship to an animal is based on trust”, was counted 36 times, the most frequently counted category after category 1 and category 2. As shown in Table 2, category 3 was mentioned by all participants. Taking the total counting number as well as the frequency for each participant into account, no other category was named as frequently as category 3 except for category 1 and category 2. Notably, category 1, category 2 and category 3 were the highest counts and delivered the most detailed narratives and consequently were processed more frequently during content analysis. 

In category 3, a multitude of different factors related to a deep trust between animals and humans were mentioned. Participants characterized animals as authentic and honest. They reported a general feeling that animals offer unconditional help. One participant remembered an AAT session in which he could fully be himself, because he did not feel judged by the animal. Moreover, participants emphasized the ability of animals to express empathy without pity. Similar to category 1, participants viewed this trusting relationship to animals in contrast to relationships with humans. 

Categories 4 and 10 focused on participants’ belief in a positive outcome of AAT. Category 4 refers to AAT as being helpful and category 10 to AAT as being effective. Helpful was preferred over effective and mentioned over 11 times more often. Participants reported that they were grateful for receiving AAT during their rehabilitation stay, and they hope that this therapy form is available for future patients. Categories 5 to 9 reflected additional explanations of why participants believe AAT is helpful, and how they understood the memory effects of AAT.

## 4. Discussion

We found that patients with acquired brain injury remembered pictures of AAT sessions significantly better than pictures of comparable standard therapy sessions that they attended two years ago. This result suggests that the presence of animals during therapy sessions might enhance episodic memory in patients with acquired brain injury in the long-term. This is in line with previously completed studies indicating that the presence of an animal contributes to an enhanced motivation and attention [14,19,27,28,29,30]. Moreover, we found a significantly higher number of positive emotions and a significantly lower arousal when patients looked at pictures from AAT sessions compared to comparable standard therapy sessions. This is also in line with previous studies indicating that animals can trigger positive emotions [14,15,16] and reflects a possible mechanism underlying this enhancement of episodic memory in the presence of an animal. Positive emotions seem to enhance neuronal plasticity and, therefore, improve episodic memory and learning [27,31]. This effect is especially high if the learning cue is emotionally meaningful, as shown in the results of the qualitative interviews describing the relationship between the patients and the involved animals. These findings are in line with results from previous studies indicating that animals positively influence emotions, learning and motivation [21,31,32]. Interestingly our results show that pictures of AAT induced lower arousal than pictures from standard therapy sessions. This is contrary to studies indicating that the presence of an animal can lead to higher arousal [14,15,17,18]. It is also surprising as previous research shows that high arousal leads to better memory performance [31]. Our results suggest that a relationship to an animal might be enough to influence the memory of the patients. Future studies should further investigate the connection between animal presence and arousal and its effects on memory performance by systematically varying arousal within the AAT sessions. In the qualitative analysis, we detected a further possible mechanism for enhanced memory in the trusting relationship to an animal. Trust has been described as an important characteristic in the therapist–patient relationship [33,34]. For example, the patient–therapist relationship and trust in the therapist seem to positively influence the therapy outcome [35]. The possibility to facilitate and strengthen the therapeutic relationship by integrating an animal into therapy might therefore be an innovative and promising approach, not just to enhance episodic memory but to facilitate the whole therapy process. Interestingly, trust seems to affect not only the therapist–patient relationship, but also seems to have a direct influence on health care outcomes [36,37,38]. For example, a recent meta-analysis from Birkhäuser and colleagues [39] found that patients reported more beneficial health behavior, fewer symptoms, a higher quality of life and were more satisfied with treatment when they had higher trust in their health care professional. These results suggest that trust in the therapist is a core element of an effective treatment. In this context, the question arises if patient trust in the present animal might have similar effects on treatment outcome and if the experienced trust in the animal has the power to positively influence the therapist–patient relationship. More research about the causality of trust and its influences on episodic memory and health outcomes are needed. Such research is not only necessary to address effectiveness of AAT but would also contribute to common factor research and investigate the interesting possibility to modify the factor of trust between patient and therapist in clinical practice through incorporating an animal. 

A major limitation of this study was the small sample size of this pilot study. We corrected for a small sample with Hedges g and still found moderate effects for episodic memory, as well as large effects for positive emotions and arousal. However, our quantitative results should be interpreted with caution. This first attempt to investigate long-term effects of AAT should be replicated with larger samples in the future. The same applies to the qualitative data that provide an insight in a small population, which should not be over interpreted. There may also be a sampling bias affecting our results, because all included patients seem to have a strong solidarity with the rehabilitation clinic, whereas the patients who were not motivated to participate in the study might have had less connection with the rehabilitation clinic and its staff. Moreover, different participants had different therapists, which might have influenced our results. However, for each participant, the same therapist performed both the sessions with and without animals. Future research should include more patients, as well as enlarge it into further populations with other clinical problems, for example dementia. Further studies should specifically control for therapists’ variables. The design in this study just allows conclusions about the episodic memory on the presence of animals but not on the content of the therapy sessions because we did not ask patients if they remember certain tasks or content of the past therapy sessions. Our qualitative data indicate that patients did remember therapy content of AAT sessions, but we could not directly evaluate if patients could remember AAT session content better than therapy content from standard therapy sessions. Moreover, the arrangement of the interview questionnaire could have led to a priming effect in participants. Talking about AAT in the interview might have affected questionnaire ratings that followed afterwards. There were more men than women participating in this study. Results cannot, therefore, be generalized. However, Pefke and colleagues [40] found no significant gender differences on the behavioral level in memory performance influenced by different emotional intensities of memories. To date, there is little research addressing the effect of gender on the perception of human-animal interaction and the effects of animal-assisted therapy. A study with a small sample size by Marr and colleagues [41] found no effect of gender investigating AAT in psychiatric rehabilitation. Other research such as a study by Stetina and colleagues [42] points at different effects of AAT in male or female inmates. Future studies should try to balance the gender ratio. Moreover, future research should combine the topic of human–animal interaction, health research and gender research. Finally, it must be kept in mind that results on episodic memory rely heavily on memory paradigms. It makes a difference if participants retrieve memories from everyday life or autobiographical memories, and if the presented cue is a word, a picture or a video [43]. For example, Chen and colleagues showed that video and picture material from autobiographical content could be remembered better than an emotional word. In our study, we used one of the most recommended paradigms to investigate episodic memory: image material of experienced events from participants’ everyday life [43]. Nevertheless, it should be emphasized that our study just allows a small insight in participants’ episodic memory. Further research is needed, preferably in an experimental setting, which allows for manipulation of the therapy content and using different methodological approaches and study designs. Furthermore, only one person collected and analyzed the data presented here. Although we tried to prevent researcher bias by discussing the methods as well as the qualitative results in an expert group including a researcher from a different field, researcher allegiance may have had an impact on the results. Thus, our results should be seen as a first insight into the effects of AAT on long-term episodic memory which generate hypotheses, such as positive emotions and trust as possible mechanism, that need to be tested in the future. 

## 5. Conclusions

The results from our pilot study provide a first indicator that AAT might enhance long-term episodic memory in patients with acquired brain injuries, possibly through enhanced positive emotions and trust in the presence of an animal. This suggests that AAT might be an important approach for neurorehabilitation in patients with acquired brain injuries. However, future research should replicate and further investigate this outcome and the underlying mechanism [44]. Based on our qualitative findings, we suggest that besides enhanced positive emotions, trust within the therapist–animal–patient triad might be an important underlying factor.

## Figures and Tables

**Table 1 ijerph-17-08466-t001:** Arousal and valence of pictures of animal-assisted therapy (AAT) and standard therapy sessions.

Parameter	Condition	M ^1^	Mdn ^2^	Sd ^3^	CI ^4^	Z	*p*	g
Arousal	no animal	3.43	3.50	1.02	2.87–4.00	−3.36	0.03	0.96
	with animal	2.10	2.00	0.78	1.67–2.53			
Valence	no animal	2.68	2.50	1.17	2.02–3.33	−2.75	0.006	0.90
	with animal	1.71	2.00	0.64	1.36–2.07			

^1^ M = mean, ^2^ Mdn = median, ^3^ sd = standard deviation, ^4^ CI =95% confidence interval.

**Table 2 ijerph-17-08466-t002:** Overview of the full category system and counting summary for category 1 to 10 from each of the interviews (I).

Category	Description	I1	I3	I4	I6	I7	I8	Total
1	Positive emotions related to animals.	13	22	3	15	7	14	**74**
2	Good memory of animals.	4	6	5	4	8	10	**37**
3	The relationship to an animal is based on trust.	2	18	2	4	8	2	**36**
4	AAT is helpful.	5	7	3	8	6	6	**35**
5	Animals enable a different connection to oneself.	3	8	-	1	-	2	**14**
6	Animals enable a different sort of learning.	1	8	4	-	-	-	**13**
7	Memories on animals are different.	1	7	-	2	3	-	**13**
8	Differences between animals and humans.	3	4	3	-	2	-	**12**
9	Differences between animals and therapists.	2	3	1	1	1	2	**10**
10	AAT is effective.	2	1	-	-	-	-	**3**
11	Animals as a good thing, which is not effective for everyone.							
12	Animals at the REHAB Basel.							
13	Highlight animal-garden.							
14	Animals as role models.							
15	Animals as part of the family.							
16	Animals as self-determinant individuals.							
17	Animals as companion and protector.							
18	Favorite animal.							
19	Animals as daily help.							
20	Pets as stress factor.							
21	Animals as integral part of our society.							
22	Hippotherapy as hard but helpful.							
23	“Exercise life” as great time.

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
