# Peer review of "Influences of Animal-Assisted Therapy on Episodic Memory in Patients with Acquired Brain Injuries"

_ijerph, 2020, doi:10.3390/ijerph17228466_

Round 1
Reviewer 1 Report
The manuscript, "Influences of animal-assisted therapy on episodic memory in patients with acquired barin injuries," presented the evaluation of therapy dogs in some sessions of an intervention for acquired brain injuries.
The authors select the patients from a previous study and explore the responses after participating in the other protocol.
The authors found a relationship between the dogs’ presence in the sessions and the elicitation of certain emotions and memoirs. My comments on this manuscript are as follows.
Overall, the abstract is readable and contains sufficient elements to know the paper’s content.
Also, the Introduction presents the rationale and conceptualization that supports the aim of the study. With a proper hypothesis.
In the method’s section, please provide details about the exclusion criteria of the participants, the therapy dog (such as breed, age, gender) and the aims of its participation: do the dog helped in certain activities? It was only present during the sessions?
In the discussion. Since the majority of participants were males. How does the gender issue impact the results? Please provide more details about the effects of arousal in this section, it is mainly focused on the valence.
Please add the hypothesis generated after reviewing your data (line 266-267).
Author Response
We thank the reviewers and the editor for the helpful comments. The changes are highlighted with track-change in the manuscript and in the following, you find our point-by-point responses.
“In the method’s section, please provide details about the exclusion criteria of the participants“.
Thank you for pointing this out. We included the exclusion criteria in the manuscript (line 62-63): “Exclusion criteria were low cognitive or speech abilities as patients needed to be able to complete the open interview or/and the written questionnaire.”
„The therapy dog (such as breed, age, gender) and the aims of its participation: do the dog helped in certain activities? It was only present during the sessions?“
Different animals took part in the study. We have included information about the present animals during AAT sessions and briefly described the certain activities during AAT sessions. A more extensive description can be found in the article of the previous study. We included the following sentences (line 84-86): “The sessions were matched within one patient for activities, goals and setting. Involved animals were guinea pigs, rabbits, chickens, miniature pigs, goats, sheep, horses, donkeys, cats and dogs. In AAT sessions, patients have been given tasks to care for or feed the animal whereas in the control sessions, similar tasks were carried out but without the presence of an animal [15]. The patients and the animals actively interacted during the AAT sessions but often only a part of the time of a whole session.”
„In the discussion. Since the majority of participants were males. How does the gender issue impact the results?”
Thanks for raising this aspect. We added a section discussing this topic (line 302-310). “There were more men than women participating in this study. Results can therefore not be generalized. However, Pefke and colleagues [40] found no significant gender differences on the behavioral level in memory performance influenced by different emotional intensities of memories. To date, there is little research addressing the effect of gender on the perception of human-animal interaction and the effects of animal-assisted therapy. A study with a small sample size by Marr and colleagues [41] found no effect of gender investigating AAT in psychiatric rehabilitation. Other research such as a study by Stetina and colleagues [42] points at different effects of AAT in male or female inmates. Future studies should try to balance the gender ratio. Moreover, future research should combine the topic of human-animal interaction, health research and gender research..”
„Please provide more details about the effects of arousal in this section, it is mainly focused on the valence.”
A section to the valence was added (line 244-251). “Interestingly our results show that pictures of AAT induced lower arousal than standard. This is contrary to studies indicating that the presence of an animal can lead to higher arousal [14,15,17,18]. It is also surprising as previous research shows that high arousal leads to better memory performance [27, 30]. Our results suggest that a relationship to an animal might be enough to influence the memory of the patients. Future studies should further investigate the connection between animal presence and arousal and its effects on memory performance by systematically varying arousal within the AAT sessions.”
“Please add the hypothesis generated after reviewing your data (line 266-267).“
We added this information here again according to your comment (line 324):“…. which generate hypotheses, such as positive emotions and trust as possible mechanism, that need to be tested in the future. "
Reviewer 2 Report
The manuscript deals with a very interesting research topic.
(1).In my opinion the introduction should be expanded
(2). I believe that the information in the appendices should be included in complementary material and translated into English.
(3). Part of the content of appendix A should be included in material and methods to improve the understanding of the manuscript.
(4). The content of appendix C must be included in the results section as a table.
(5). I think it is convenient to include the limitations of the study
Author Response
We thank the reviewers and the editor for the helpful comments. The changes are highlighted with track-change in the manuscript and in the following, you find our point-by-point responses.
1) In my opinion the introduction should be expanded
We are happy to expand the introduction further but we would need to know what topic you want to be extended. We included all relevant aspects leading to the study question in a highly concise way and therefore provide more insights in the discussion section.
2) I believe that the information in the appendices should be included in complementary material and translated into English.
We agree with you. We have translated the contents of the appendix in English. Moreover, the appendix is now called complementary material.
3) Part of the content of appendix A should be included in material and methods to improve the understanding of the manuscript.
We described the questions in more detail within the text in lines 183-185. Moreover, the whole questionnaire is now translated in English and can be found in the complementary material.
4) The content of appendix C must be included in the results section as a table.
Thank you for this suggestion. We included the categories with all the collected data in the results section in table 2.
5) I think it is convenient to include the limitations of the study
We are not quite sure to what this comment refers. In the paper, limitations of the study are discussed and clearly stated as limitations. They are discussed together with suggestions for future research from line 291 to line 377. If there is something more that you would like us to include, we are happy to.